# Revealing Allosteric Mechanism of Amino Acid Binding Proteins from Open to Closed State

**DOI:** 10.3390/molecules28207139

**Published:** 2023-10-17

**Authors:** Quanshan Shi, Ling Liu, Huaichuan Duan, Yu Jiang, Wenqin Luo, Guangzhou Sun, Yutong Ge, Li Liang, Wei Liu, Hubing Shi, Jianping Hu

**Affiliations:** 1Key Laboratory of Medicinal and Edible Plants Resources Development of Sichuan Education Department, School of Pharmacy, Chengdu University, Chengdu 610106, China; sqs_715@163.com (Q.S.); yzh2602023@163.com (L.L.); jy94940514@163.com (Y.J.); m13568453654@163.com (W.L.); s19196482291@163.com (G.S.); 13111153368@163.com (Y.G.); liangli_cdu@163.com (L.L.); liuwei@cdu.edu.cn (W.L.); 2Laboratory of Integrative Medicine, Clinical Research Center for Breast, State Key Laboratory of Biotherapy, West China Hospital, Sichuan University and Collaborative Innovation Center, Chengdu 610041, China; sumdhc@163.com

**Keywords:** amino acid binding protein, allosteric mechanism, Gaussian network model, anisotropic network model, neural relational inference molecular dynamics

## Abstract

Amino acid binding proteins (AABPs) undergo significant conformational closure in the periplasmic space of Gram-negative bacteria, tightly binding specific amino acid substrates and then initiating transmembrane transport of nutrients. Nevertheless, the possible closure mechanisms after substrate binding, especially long-range signaling, remain unknown. Taking three typical AABPs—glutamine binding protein (GlnBP), histidine binding protein (HisJ) and lysine/arginine/ornithine binding protein (LAOBP) in *Escherichia coli* (*E. coli*)—as research subjects, a series of theoretical studies including sequence alignment, Gaussian network model (GNM), anisotropic network model (ANM), conventional molecular dynamics (cMD) and neural relational inference molecular dynamics (NRI-MD) simulations were carried out. Sequence alignment showed that GlnBP, HisJ and LAOBP have high structural similarity. According to the results of the GNM and ANM, AABPs’ Index Finger and Thumb domains exhibit closed motion tendencies that contribute to substrate capture and stable binding. Based on cMD trajectories, the Index Finger domain, especially the I-Loop region, exhibits high molecular flexibility, with residues 11 and 117 both being potentially key residues for receptor–ligand recognition and initiation of receptor allostery. Finally, the signaling pathway of AABPs’ conformational closure was revealed by NRI-MD training and trajectory reconstruction. This work not only provides a complete picture of AABPs’ recognition mechanism and possible conformational closure, but also aids subsequent structure-based design of small-molecule oncology drugs.

## 1. Introduction

Gram-negative bacteria are similar to chloroplasts and mitochondria organelles, containing an inner and outer bilayer membrane with a periplasmic space between them [1]. The outer membrane consists of two layers of phospholipids and embedded proteins which play an important role in maintaining structural stability of the bacterium, protecting the internal organelles and responding to environmental stress. Polysaccharide molecules are involved in forming the outer layer of the outer membrane, acting as a protective shield against harmful substances. In addition, the channel proteins in the outer membrane dominate the formation of pores, regulating the entry of hydrophilic substrates into the periplasmic space [2]. Periplasmic binding proteins (PBPs) unique to periplasmic space can interact with channel proteins in the outer membrane to promote the absorption and transport of exogenous small molecules, together forming the ATP-binding cassette (ABC) transporter system. Notably, PBPs are highly specific to the substrate, enabling the bacteria to take up critical nutrients in a survival environment [3,4]. In a word, PBPs play an essential role in the nutrient transport system of Gram-negative bacteria, as well as in maintaining bacterial survival, adaptation and pathogenic resistance.

PBPs were first identified by Neu et al., during cold osmotic shock treatment of *Escherichia coli* (*E. coli*) K12. Proteins with amino acid binding activity were found in periplasmic membrane contents of damaged cells [5]. Since then, more PBPs from Gram-negative bacteria, including *E. coli* and *Salmonella typhimurium* (*S. typhimurium*), have been isolated and structurally characterized. These PBPs transport a variety of substrates, such as amino acids, sugars, peptides, anions and vitamins [6]. In the *E. coli* system, amino acid binding proteins (AABPs) specifically recognize amino acids [7] showing common properties: monomeric proteins, molecular weight of 22–59 kDa and thermal stability [8]. Currently, widely studied AABPs are glutamine binding protein (GlnBP) [9], histidine binding protein (HisJ) [10], lysine/arginine/ornithine binding protein (LAOBP) [11], leucine/isoleucine/valine binding protein (LIVBP) [12] and leucine-specific binding protein (LSBP) [13]. Among them, GlnBP, HisJ and LAOBP are all from *E. coli*, while LIVBP and LSBP are found in Klebsiella pneumoniae and Salmonella enterica, respectively. They are highly similar in structure and function, including hinges and large and small domains, with the substrate binding site being located in the pocket formed between two domains [14,15]. Based on crystallographic studies of pocket shape, two homeostatic conformational states were identified: the substrate-free open-cleft and substrate-bound closed-cleft states, which dominate the binding and release of exogenous amino acids or peptides, respectively [16]. When the amino acid or peptide substrates required by the bacterium are present in the periplasmic space, AABPs in the open state actively recognize the substrate specificity, accompanied by conformational closure. The substrate-carrying AABPs are then recognized by the corresponding receptor on the cytoplasmic membrane to complete substrate transport and release. Currently, the GlnBP-dominated Gln transport system is one of the most thoroughly studied permease transport systems relying on binding proteins. Through genetic and mutation experiments, Weiner et al., and Masters et al., found that the transport of Gln by mutated GlnBP was drastically reduced, instead relying on osmotic pressure [17,18]. Sun et al., assessed the crystal structures of the GlnBP monomer (2.30 Å) and complex with Gln (1.94 Å) by X-ray in 1996 and 1998, respectively. GlnBP contains 226 amino acids which consist of the large (i.e., Thumb domain) and small (i.e., Index Finger domain) globular domains with similar secondary structures connected by two Linkers [15,19]. HisJ and LAOBP both are composed of 238 residues and have a 3D structure similar to GlnBP. The three systems of GlnBP, HisJ and LAOBP bind substrates as if grasping a ball by hand, that is, with Linkers as the axis; Thumb and Index Finger domains are closed to each other to catch its specific substrate (see Figure 1B). Unlike GlnBP/HisJ/LAOBP, LIVBP and LSBP systems are larger overall, consisting of 344 and 346 residues, respectively.

Allosteric effects of AABPs can lead to changes in protein active sites or binding pockets which, in turn, affect molecular recognition by specific substrates. In recent years, the allosteric effect of the GlnBP, HisJ and LAOBP systems has become a hot scientific issue. In 2004, Sun et al., performed molecular dynamics (MD) simulations to explore motion patterns of GlnBP, revealing the balance between open and closed states. Specifically, the GlnBP monomer exhibits more obvious open–closed motions than the GlnBP–Gln complex, and residues A96-K110 in both systems exhibit higher molecular flexibility [20]. Using a simple spring model instead of an all-atom MD simulation, Su et al., found that the topology and inter-domain interactions both determine functional motion of the GlnBP system [21]. Focusing on important scientific issues, including conformational dynamics and signaling mechanism, Chu et al., carried out experiments based on catalytic activity, binding affinity and nuclear magnetic resonance (NMR) spectroscopy for the HisJ recombinant protein [22]. The results indicate that HisJ tends to adopt the open state in the absence of a ligand and then transforms to a closed state after binding the substrate, driven by the motion of two hinge Linkers [22]. According to MD trajectories, Chu et al., found that HisJ binding to the substrate mainly remains in a closed state accompanied by rotational motion, which can also feedback and affect inter-domain interactions [23]. By computer-aided molecular simulation, MD simulation and protein engineering, Banda-Vazquez et al., successfully reconstructed LAOBP, which also has open and closed states, into an amino acid sensor with broader species adaptability [24].

Billions of years of evolution have provided organisms with a complex network of molecular interactions, and allosteric effects allow them to respond efficiently to environmental changes. Current experimental strategies to explore allosteric effects include: X-ray crystallography, nuclear magnetic resonance (NMR) and cryo-electron microscopy (cryo-EM), determining the 3D structure of biomolecules in different states and extracting information on conformational changes; time-resolved fluoroimmunoassay (TRFIA), amplifying specific recognition signals to determine substrate concentrations and infer their possible molecular structures; and isothermal titration calorimetry (ITC), providing thermodynamic and kinetic information by monitoring and recording the calorimetric profile of the receptor–ligand molecule recognition process. With the optimization of computational methods, numerous molecular simulation methods can be used to capture the conformational characteristics of biological macromolecules at the atomic level and on different time scales. The Gaussian network model (GNM) can extract the flexible distribution of fast/slow motions of biomolecules, folding cores and motion correlations, etc. [25,26]. The anisotropic network model (ANM) can provide the magnitude and absolute direction of allosteric motion [27]. MD simulations capture a variety of critical biomolecular processes, including ligand binding, protein folding, allosteric regulation and catalytic activation of enzymes. By combining deep learning strategy neural relational inference (NRI) with the MD simulation method, long-range allosteric effects and signaling pathways of proteins both can be revealed. In fact, neural relational inference molecular dynamics (NRI-MD) simulations have been successfully applied to explore core bioscience problems such as protein folding, enzyme activation and receptor–ligand recognition [28].

The conformational conversion from open to closed states shows the same characteristics for the GlnBP, HisJ and LAOBP systems from *E. coli*, i.e., with the binding to substrates, their large/small structural domains are close to each other, accompanied by the weakening of inter-domain motion. Currently, several scientific questions remain unclear. Why are these three AABPs able to bind to different substrates despite possessing similar structure? How do the interactions between protein residues change during the allosteric process of AABPs? In addition, which residues are involved in the allosteric signaling transduction network? In this work, the allosteric effects of three AABPs’ binding substrates from open to closed states were investigated from both global and local perspectives via GNM, ANM, MD simulation and NRI-MD strategies. It not only helps to understand substrate capture and conformational change characteristics of AABPs, but also provides valuable theoretical information for substrate transport and remote signaling.

## 2. Results and Discussion

### 2.1. GlnBP, HisJ and LAOBP Showing High Structural Similarity

Gram-negative bacteria AABPs play an important role in amino acid metabolism and nutrient transport, and their structure and function have received extensive attention [29,30]. In this work, GlnBP, HisJ and LAOBP were selected as the research systems for the following two reasons: (1) they are all from *E. coli*, which are the most commonly used model microorganisms in life science research, and have complete genomic information; (2) their total residue numbers are 226, 238 and 238, respectively, and the similar chain length is more conductive to difference comparison in conformational changes. In order to determine the similarity among GlnBP, HisJ and LAOBP, Figure 2 shows their primary sequences and secondary and tertiary structures. As shown in Figure 2A, the sequence coverage of the three systems is above 95%. Specifically, HisJ and LAOBP have 70.46% sequence identity; the sequence identity of GlnBP with HisJ/LAOBP is slightly lower, respectively maintaining 28.38% and 30.67%, which may be related to the five-amino-acid gap appearing after GlnBP A187. The 52 highly conserved residues in the three systems may be deeply involved in structural stability and substrate transport of AABPs [31].

The GlnBP, HisJ and LAOBP systems also show highly similar 3D structure, with only residues 66–69, 86–90, 106–109 and 193–196 differing significantly (see black box in Figure 2A). According to 3D structure superimposition of GlnBP-open/HisJ-open/LAOBP-open, the Thumb is distributed in residues 1–85 and 197–238 and the Index Finger is located in residues 91–192, while the Linker is composed of residues 86–90 and 193–196 (see Figure 2B). Comparing Figure 2A,B, the two regions with large structural differences (i.e., residues 86–90 and 193–196) are located in the Linker, where GlnBP is a β-sheet, while HisJ and LAOBP both are random loops. Overall, the high similarity of sequences and structures indicates the close functional evolution of GlnBP/HisJ/LAOBP, while the gaps in sequence alignment and the differences in four secondary structures both determine the subtle differences in their 3D structures and substrate recognition specificity.

### 2.2. Allosteric Effects Revealed by Elastic Network Models

Under the influence of mutations or external factors (e.g., binding to substrates or exposure to different pH), proteins may undergo significant conformational changes known as allosteric effects. It not only affects the stability and biological function of the system, but is also closely related to the occurrence of diseases and drug action [30,32,33].
Fast Motion Patterns Revealing Topological Characteristics of AABPs

Figure 3A,B show conformational fluctuation of fast motion for the open and closed states of GlnBP/HisJ/LAOBP according to Gaussian network model (GNM) analyses, respectively. The flexible distributions of the six systems are close, with a correlation coefficient over 0.84, which indicates that the topological connections and local structures both are similar. By comparing fast motion changes of the GlnBP/HisJ/LAOBP systems after binding the substrates, the internal domains of the Index Finger and Thumb are highly coupled, maintaining independent rigid characteristics. In general, hot-spot residues in fast motion make an important contribution to maintaining protein stability or substrate recognition. As shown in Figure 3B, the hot-spot residues of the GlnBP-closed, HisJ-closed and LAOBP-closed systems include residues 18 and 116, corresponding to E17/V114 for GlnBP, E18/V116 for HisJ and S18/V116 for LAOBP. In fact, residue 18 is located in the outermost exposed region, suggesting that HisJ E18 and LAOBP S18 both are similar to GlnBP E17 and may be involved in binding to membrane receptors [19]. In addition, residue 116 (i.e., GlnBP V114, HisJ/LAOBP V116) is located in the active pocket and dominates substrate recognition by maintaining the overall rigidity of the system [34,35,36].

Flexible Distribution of AABPs Slow Motion

Figure 3C,D show slow motion modes for the open and closed states of GlnBP/HisJ/LAOBP. The conformational fluctuations of the three systems are very similar, with the Linker visually and effectively distinguishing their large/small domains. Interestingly, the conformational fluctuations of residues 89 and 193 in the Linker region are close to zero, which means that it is the Index finger and Thumb regions, rather than the more rigid Linker, that drive AABPs’ conformational changes. As shown in Figure 3C, when the protein remains in the open state, the Index Finger region exhibits higher flexibility than the Thumb region, especially for residues 96–150, including two α-helices and two β-sheets. It suggests that the Index Finger domain plays a central role in the transition of AABPs from the open to closed states [13]. As shown in Figure 3D, the overall molecular flexibility of AABPs decreases after binding the substrate. Unexpectedly, residues 221–238 at the C-terminal tail showed greater flexibility, and it did not participate in or affect the conformation of closed–open state transition. The conformational fluctuation of the N-/C-terminal in the AABPs–substrate complex decreased sharply, corresponding to tight association of receptor–ligand and lower molecular flexibility, which is consistent with the results of Su et al.’s elastic network model (ENM) [21].Motion Correlation of AABPs

Figure 4 shows motion correlations for the investigated AABPs’ monomer and complex systems (i.e., the GlnBP-open, GlnBP-closed, HisJ-open, HisJ-closed, LAOBP-open and LAOBP-closed states). On the whole, whether in the open or closed state, the three AABPs show similar motion correlation distribution, mainly composed of five warm and four cool color regions. For the three AABPs’ open states, the Index Finger possesses highly negative motion correlation with that of the Thumb (indicated by four cold color regions), exhibiting significant open–closed motions. According to the five warm color zones, residues in the Index Finger moved in a highly coupled manner, while the Thumb showed relatively weak positive correlation. Moreover, the warm color in the open state is darker than that in the closed state, indicating that the association with substrates will strengthen the motion integrity of AABPs’ large/small domains and weaken their functional partitioning.

The motion patterns of AABPs contributing to capture of amino acid substrates

The above correlation analysis only provides the relationship between residue motion directions. In fact, the determination of absolute motion direction of the investigated system is a prerequisite for an in-depth understanding of its biological function. Figure 5 shows functional motion patterns for the open and closed states of GlnBP/HisJ/LAOBP via ANM analysis. As shown in Figure 5, the motion patterns are similar for the three open states. The motion range of the Index Finger is slightly larger than that of the Thumb domain, and their face-to-face movement causes the pocket to close, which contributes to both capturing of specific amino acid substrates and stability of the complex. It can also be seen that the closed motion is dominated by the large/small domains rather than the Linker, which also confirms the above slow motion analysis results. In addition, with the association of amino acid substrates, AABPs are relatively rigid on the whole, except for some loop regions.

### 2.3. Interaction of Protein Fragments Inferred from Neural Relational Inference Molecular Dynamics

Above, two elastic network models were adopted to explore conformational changes of the GlnBP, HisJ and LAOBP systems. In particular, the GNM results indicate that AABPs share the same hot-spot residues and functional motion regions. According to ANM analysis, AABPs’ monomers exhibit relatively high mobility and flexibility. The motion mode is similar to that of grasping objects by hand, that is, specific amino acid substrates are captured by face-to-face motion between Index Finger and Thumb, with domain flexibility ranked from high to low as follows: Index Finger, Thumb and Linker. When the amino acid substrate was bound by AABPs, the overall motion amplitude of the latter decreased significantly. Subsequently, the above complex is recognized by receptors on the inner membrane, accompanied by the AABPs’ pocket opening to release the amino acid substrate and cross the inner membrane. Nevertheless, how domain interactions change during AABPs’ pocket closure and what the signaling pathway at the amino acid level is both have not been reported. In this work, the NRI-MD strategy, which combines the graph neural network and cMD, is used to explore the above two important scientific problems [28,37].

Convergent cMD simulations are a prerequisite for NRI-MD training

Before the NRI-MD training, cMD simulations of 300 ns are performed for the six systems (i.e., GlnBP-open, GlnBP-closed, HisJ-open, HisJ-closed, LAOBP-open and LAOBP-closed). Figure 6 shows the convergence analysis results of cMD simulation trajectories. In terms of the root-mean-standard deviation (RMSD) of all C_α_ atoms, the three AABPs’ systems are similar in general, with the open state being slightly larger than the closed state and showing stronger mobility. Both the low absolute value and stable RMSD distribution fully prove the reliability of the cMD simulation process, which can be used for subsequent NRI model training and trajectory reconstruction.

Figure 6B,C show the root-mean-standard fluctuation (RMSF) values at the residual level, as well as its correlations between the AABPs’ closed and open states. The RMSF parameter can be used to reflect the flexibility distribution of a protein system. The higher the RMSF value, the more flexible the corresponding region is, and the protein rigid region is the opposite. On the whole, the six systems show similar RMSF distribution. Specifically, the RMSF values of the three closed states (i.e., GlnBP-closed, HisJ-closed, LAOBP-closed) are lower than those of the corresponding open states (i.e., GlnBP-open, HisJ-open, LAOBP-open), indicating that the AABPs’ structure in the closed state is more stable. As mentioned earlier, AABPs are separated by a Linker between two structural domains (i.e., Index Finger and Thumb). Here, the RMSF values around residues 90 and 193 in the Linker are less than 0.1 nm, remaining relatively rigid. In the AABPs’ closed states, residues with RMSF values greater than 0.18 nm are mainly distributed in the Index Finger domain, exhibiting stronger motility. The RMSF values of the GlnBP/HisJ/LAOBP open states have a significant correlation with those of their closed states, remaining at 0.85/0.71/0.86, respectively, which, again, fully proves the convergence and reliability of the six cMD trajectories. In addition, all the above cMD results are consistent with the data from the coarse-grained models.

Rational domain partitions for subsequent NRI-MD analysis

As shown in Figure 2B, the three investigated AABPs (i.e., GlnBP, HisJ and LAOBP) have similar secondary structures overall. To further investigate the interactions between intramolecular domains, AABPs can be subdivided into eight fragment parts (see Figure 7). Specifically, T-α (residues 1–17), IT1 (residues 61–91), I-βα (residues 113–156) and I-2βα (residues 157–185), showing a tendency for pocket closure, are all involved in molecular recognition and anchoring with amino acid substrates [19]. T-αβα (residues 18–60) is evolutionarily conserved and has relatively weak motion character, consisting of two α-helices and one β-sheet. Located in the terminal of the Index Finger, I-β (residues 92–97) connects to the Linker and participates in the pocket closure, being prone to partial torsion [36]. The I-Loop (residues 98–112) in the Index Finger also shows high flexibility, contributing to the closure of the substrate recognition pocket [20]. AABPs in the closed state are generally rigid, with the exception of IT2 (residues 186–238), which is closely related to both association with the receptor on the inner membrane and substrate release [38].

Domain communications of three AABPs’ systems

Based on reliable cMD trajectories and rational domain partitions, the domain interactions for AABPs’ allosteric communication are subsequently investigated. On the basis of cMD trajectories, the NRI model is trained with the adjacent residues being integrated as blocks. Figure 8A,B show the distribution of residue interactions among learned edges and domains for six AABPs’ systems (i.e., GlnBP-open, GlnBP-closed, HisJ-open, HisJ-closed, LAOBP-open). The learned edges of AABPs’ open states appear mainly between the other structural domains and I-β (residues 92–97)/I-Loop (residues 98–112)/IT2 (residues 186–238), suggesting that the Index Finger and Thumb terminals both receive the allosteric signal. As shown in Figure 8B,C, the domain interactions of the three AABPs are highly similar. In the AABPs’ open states, almost all domains interact with the I-Loop (residues 98–112). In the AABPs’ closed states, the effect of the Index Finger is greatly weakened, while that of the Thumb and I-Loop is still maintained. Since the global cooperative motion is concentrated in the I-Loop (98–112), it is presumed to be the key to the allosteric process of AABPs.

### 2.4. Communication Pathways in the Allosteric Process of AABPs


Key residues for receptor–ligand recognition


The conversion in AABPs from the open to closed state is caused by amino acid substrate binding, and thus it is reasonable to infer that key residues for receptor–ligand recognition may be the starting point for transmission of allosteric communication. On the basis of cMD trajectories, Table 1 lists the key residues with binding energy lower than −1.00 kJ/mol, calculated with the MM-GBSA energy decomposition method [39], which are conducive to the recognition of GlnBP/HisJ/LAOBP by their specific amino acid substrates Gln/His/Lys. Overall, residues 11 and 117 (i.e., D10/K115 in GlnBP, D11/L117 in HisJ and LAOBP) both contribute to molecular recognition of AABPs by specific amino acid substrates [20]. There are partially consistent key residues, such as GlnBP A67 vs. HisJ S69, GlnBP G68 vs. LAOBP S70, HisJ L52 vs. LAOBP F52. In addition, there are unique key residues involved in substrate recognition. For example, the residues G119 and D157 in GlnBP form hydrogen bonds with the substrate Gln [36]; the aromatic side-chain of Y14 in HisJ interacts tightly with the substrate His, undergoing significant conformational changes [34]; R77 charged side-chain and S72 side-chain hydroxyl group both contribute to the stable association of LAOBP with substrate Lys [40]. It is speculated that the identical key residues may lead to conformational closure of AABPs, while the unique residues may be responsible for the specific recognition of substrates.


Signal transmission pathways for conformational closure of AABPs


Key residues contributing to substrate recognition are defined as signal starting points for inducing conformational changes; next, the residues in the I-Loop with the highest flexibility are set as signal endpoints, and then the shortest pathways between amino acids for remote signal transmission are subsequently explored. Appendix A lists all possible shortest pathways of signal transmission in AABPs to the region near the I-Loop (residues 97–111). It can be seen that there is some overlap and crossover of signal pathways, which is conducive to the enhancement of allosteric signals and feedback regulation. The frequency of occurrence for each residue in the pathway can be used to characterize its relative importance in global connectivity and mediating potency, where high-frequency residues contribute to the enhancement of allosteric signals from activation. Figure 9 shows the most likely pathways mediating inter-domain allosteric communications in AABPs, all of which start from residue 117. Although the participating residues in the signaling pathways of the three AABPs’ open systems are different, the signals all originate from the Index Finger region with strong motility, which is consistent with the previous GNM/ANM/cMD results. After AABPs are bound by specific amino acid substrates, the intramolecular interactions become more compact, with signaling being more complex, which factually facilitates subsequent recognition by receptors on the inner membrane to complete the release of substrates. As a common key residue for substrate recognition, residue 117 has the potential to induce pocket closure of AABPs.

### 2.5. Possible Allosteric Mechanism of AABPs’ Binding Substrate

With reference to the above series of molecular simulation data, the pocket closure effect of three representative *E. coli* AABPs’ binding substrates was investigated in detail. Specifically, the allosteric trend of AABPs was predicted based on GNM and ANM analyses. The interactions between structural domains, as well as possible signaling pathways of AABPs, are revealed from the NRI-MD results. A possible allosteric mechanism of AABPs was proposed by combining global/local conformational changes and molecular recognition analysis (see Figure 10). The specific amino acid substrate first passes through channel proteins on the outer membrane of *E. coli* and is recognized by key residues in the AABPs’ active pocket (see Table 1). Then, residues 11/117 (D10/K115 in GlnBP-closed, D11/L117 in HisJ-closed and LAOBP-closed) initiate the allosteric signal transmission, which eventually reaches the I-Loop, passing through the Index Finger structural domain (see Figure 9 and Figure 10B, Appendix A). Notably, the Index Finger has a stronger allosteric and substrate binding ability than the Thumb, the I-Loop possesses a high molecular flexibility and the AABPs’ open topology contains the potential for closed motion, which all aid the substrate to be tightly identified and bound (see Figure 5).

The allosteric effect of AABPs binding to the substrate and the signal transduction pathway both are proposed in this work, where the simulated results agree well with the experimental data. It is worth mentioning that a variety of biological experiments—including circular dichroism (CD) for predicting secondary structure, nuclear magnetic resonance (NMR) for providing 3D multiple conformations in solutions and site-specific mutations—are also recommended to verify the above molecular simulation results.

## 3. Materials and Methods

### 3.1. Preparation of Simulation Systems

AABPs’ monomers exhibit the open state, which is converted to the closed state upon association with corresponding substrates, and are subsequently recognized by their specific receptors in the cell membrane. The conformational transition of AABPs from open to closed state is the prerequisite for amino acid substrate recognition, transport and release. In this work, six AABPs from *E. coli* were identified as the investigated subjects—specifically, GlnBP monomer/complex, HisJ monomer/complex and LAOBP monomer/complex—whose structures were obtained from the RCSB PDB database (PDB IDs: 1GGG/1WDN, 2M8C/1HPB, 2LAO/1LST). For the convenience of analysis, the above six systems are respectively named GlnBP-open, GlnBP-closed, HisJ-open, HisJ-closed, LAOBP-open and LAOBP-closed after fully considering the state of the substrate-binding pocket.

### 3.2. Multiple Sequence Alignment

Multiple sequence alignment (MSA) can be used to predict spatial structure, folding types, biological functions and biological evolution among homologous proteins by comparing phylogenetically related protein sequences [41]. The procedure of MSA can be roughly divided into two steps: (1) numerous protein sequences are collected by searching various databases and then saved into standard FASTA format files; (2) the Basic Local Alignment Search Tool (BLAST) and Clustal Omega online servers both are used to compare the target sequences, obtaining the similarity of sequences and structures between proteins [42,43]. In this work, FASTA sequence files of GlnBP, HisJ and LAOBP were first collected from the National Center for Biotechnology Information (https://www.ncbi.nlm.nih.gov, accessed on 25 September 2023) database, and then MSA was performed on them using the BLAST server (http://www.ncbi.nlm.nih.gov/BLAST/, accessed on 25 September 2023) [44].

### 3.3. Gaussian Network Model (GNM)

In the GNM, each amino acid of a protein is represented by a point (often Cα) and is subsequently connected as a network node to form an elastic network model. It assumes that the fluctuations of all amino acids with respect to their equilibrium positions satisfy a Gaussian distribution. When the distance between two nodes is below the preset cut-off radius [45], they are connected by a spring with a fixed elastic coefficient. Compared with the traditional normal mode analysis (NMA), the GNM potential function adopts the covibrational potential instead of the conventional molecular mechanics potential function. In addition, the topology of the network can be written as an N × N Kirchhoff matrix (Γ), whose elements can be expressed by Formula (1):(1)$$\Gamma_{ij}=\left\{\begin{array}{c} -1\;\;\;\;if\;\;\;\;i\neq j\;\;\;\;and\;\;\;\;R_{ij}\leq r_{c}^{GNM}\\ 0\;\;\;\;\;\;if\;\;\;\;i\neq j\;\;\;\;and\;\;\;\;R_{ij}>r_{c}^{GNM}\\ -\sum_{i,\;\;\;\;j\neq j}\;\;\;\;if\;i\neq j \end{array}\right.$$
where $R_{ij}$ represents the distance between the *i*-th and *j*-th C_α_ atoms, and $r_{c}^{GNM}$ denotes the cut-off radius with a default value of 0.73 nm in this work. The cross-correlation fluctuation of the node *i* with *j* is proportional to non-diagonal elements in the inversion of the Kirchhoff matrix, which can be expressed by Formula (2):(2)$$\langle {∆R}_{i}\cdot R_{j}\rangle =\frac{3k_{B}T}{\gamma }\left[\Gamma^{-1}\right]$$
in which $k_{B}$ denotes the Boltzmann constant, *T* stands for the absolute temperature and γ is the elastic coefficient. The positive/negative correlation values indicate that there are similar/opposite directions of motion between residue pairs, respectively. The greater the absolute values, the stronger the motion correlation between residue pairs, while the zero value indicates that the motion is completely unrelated.

### 3.4. Anisotropic Network Model (ANM)

In the GNM, residue motion is assumed to be isotropic, and thus there is a disadvantage as only motion amplitude without direction can be provided. As an improvement, the motion pattern of proteins in the traditional ANM is characterized by the 3*N* × 3*N* Hessian matrix (*H*) instead of the previous Γ matrix in the GNM. The H-matrix can be written as:(3)$$H=\left(\begin{array}{ccc} \begin{array}{cc} h_{11} & h_{12} \end{array} & \begin{array}{cc} h_{13} & \cdots \end{array} & h_{1N}\\ \begin{array}{c} \begin{array}{cc} h_{21} & h_{22}\\ h_{31} & h_{32} \end{array}\\ \begin{array}{cc} \vdots & \;\vdots \end{array} \end{array} & \begin{array}{c} \begin{array}{cc} \begin{array}{c} h_{23}\\ h_{33} \end{array} & \begin{array}{c} \cdots \\ \cdots \end{array} \end{array}\\ \begin{array}{cc} \vdots & \;\vdots \end{array} \end{array} & \begin{array}{c} \begin{array}{c} h_{2N}\\ h_{3N} \end{array}\\ \vdots \end{array}\\ \begin{array}{cc} h_{N1} & h_{N2} \end{array} & \begin{array}{cc} h_{N3} & \cdots \end{array} & h_{NN} \end{array}\right)$$
where the elements ($h_{ij}$) of H-matrix are 3 × 3 matrices, calculated from the second-order derivatives of the potential energy (see Formula (4)):(4)$$h_{ij}=\left(\begin{array}{c} \frac{\delta^{2}V}{\delta x_{i}\delta x_{j}}\;\frac{\delta^{2}V}{\delta x_{i}\delta y_{j}}\;\frac{\delta^{2}V}{\delta x_{i}\delta z_{j}}\\ \frac{\delta^{2}V}{\delta y_{i}\delta x_{j}}\;\frac{\delta^{2}V}{\delta y_{i}\delta y_{j}}\;\frac{\delta^{2}V}{\delta y_{i}\delta z_{j}}\\ \frac{\delta^{2}V}{\delta z_{i}\delta x_{j}}\;\frac{\delta^{2}V}{\delta z_{i}\delta y_{j}}\;\frac{\delta^{2}V}{\delta z_{i}\delta z_{j}} \end{array}\right)$$

Similar to the GNM, the ANM also assumes that residue C_α_ elements are connected by springs, ignoring other effects such as hydrogen bonding on the protein molecular surface. The difference is that the ANM can provide the amplitude and direction of each residue in different motion modes and has been successfully applied for determination of key residues and exploration of the allosteric mechanism [46].

### 3.5. Molecular Dynamics Simulation

Based on Newtonian mechanics and statistical theory, molecular dynamics (MD) simulations provide theoretical predictions of thermodynamic and kinetic properties of biological macromolecules at the atomic level under random initial conditions and molecular force field potential functions. Six 300 ns comparative MD simulations were performed for the GlnBP/HisJ/LAOBP systems with open or closed pockets (respectively indicated by GlnBP-open, GlnBP-closed, HisJ-open, HisJ-closed, LAOBP-open, LAOBP-closed) via the AMBER 20 package and ff14SB force field [47]. The solute was placed in an octahedral water box with a boundary of 15.0 Å, where the solvent effect was described by the TIP3P water model [48]. A total of 16080/15162, 14896/13243 and 14473/13639 water molecules were respectively added to the GlnBP-open/GlnBP-closed, HisJ-open/HisJ-closed, LAOBP-open/LAOBP-closed systems, along with 2/2, 4/3, 2/2 Na^+^ ions. Furthermore, periodic boundary conditions (PBC) were also introduced to eliminate the boundary effects.

Prior to the MD simulations, a two-step energy optimization was performed to reduce unreasonable atomic collisions in the system: (1) 5000-step steepest descent (CD) and 5000-step conjugate gradient (CG) energy optimization at the solute-constrained state with constraint constant of 2.09 × 10^5^ kJ mol^−1^ nm^−2^; (2) solute-unconstrained optimization also consisting of the above two strategies. The convergence criterion is that the energy difference between adjacent conformations is less than 1 × 10^−4^ kcal mol^−1^ nm^−2^. After the energy minimization, two 300 ns MD simulations were then performed: (1) a 5 ns solute-constrained step where the system temperature gradually increased from 0 to 300 K with a constraint constant of 1 × 10^−3^ kcal mol^−1^ nm^−2^; (2) a constant temperature 295 ns solute-unconstrained step where the SHAKE algorithm [49] was used to prevent the breaking of all non-hydrogen chemical bonds. Throughout the MD simulations, the cut-off value of non-bonded interactions was 1 nm, and the integration step was set as 2 fs. Conformational snapshots were sampled every 500 steps (i.e., 1 ps) for each system, and thus 30,000 conformations were collected for subsequent statistical analysis, with motion characteristics monitored by VMD 1.9.3 package [50].

### 3.6. Neural Relational Inference Molecular Dynamics (NRI-MD)

The NRI-MD model has been successfully used to reveal possible remote signaling pathways by monitoring residue interactions, providing a new idea for protein design [28]. The process of NRI-MD consists of 4 main steps: (1) To prepare systems for comparison study, conventional MD simulations are carried out to extract structural information and dynamical features, which are converted into neural network format and processed to obtain the necessary statistical information; (2) Performing model training and prediction with a graphical neural network model. Specifically, input *N* amino acid nodes, and set the input/output dimension of each node *i* to 6. The eigenvectors (i.e., corresponding to the position and velocity in *x*, *y* and *z* dimensions) at time t are denoted as $x_{i}^{t}$; the eigenvector set of *N* nodes is noted as $x^{t}=\left\{x_{1}^{t},\cdots ,x_{N}^{t}\right\}$; and the trajectory for the node *i* is recorded as $x_{i}=\left\{x_{i}^{1},\cdots ,x_{i}^{T}\right\}$, where *T* is the time step. Based on the edge value learning via the NRI model, the future trajectory of the dynamic system can be reconstructed in an unsupervised manner. Edges in the network are used to represent distance relationships between nodes *i* and *j* or other specific physical/chemical relationships, $z_{i,j}\in \left\{1,\ldots ,K\right\}$, where *K* is the number of interaction types modeled. In addition, the NRI model employs a variational autoencoder (VAE) [51], which maximizes the evidence lower bound (ELBO):(5)$$L\left(\Phi ,\theta \right)=E_{q_{\Phi \left(z|x\right)}}\left[\log p_{\theta }\left(x|z\right)\right]-KL\left[q_{\Phi }\left(z|x\right)||p_{\theta }\left(z\right)\right]$$
in which Φ and *θ* both are trainable probability distribution parameters; *Z* is the factorization distribution; $q_{\Phi }\left(z|x\right)$, $p_{\theta }\left(x|z\right)$ and $p_{\theta }\left(z\right)$ are the encoder probability, decoder probability and prior probability, respectively; *K* indicates one-hot encoding with a default value of 4; (3) Selecting small batches of trajectory data for neural network training and optimizing the weights as well as parameters; it helps improve prediction accuracy for interactions between different residues; (4) By visualizing the output results of the neural network and comparing them with the statistical data of conventional MD simulations, the key residues, the shortest paths of amino acids and their correlations are identified, which eventually favors revelation of a possible protein long-range allosteric mechanism [52].

## 4. Conclusions

In this work, the conformational closure process of AABPs after binding substrates is explored by two elastic network models (i.e., the GNM and ANM) and the NRI-MD strategy embedding with GNN. The high similarity of GlnBP/HisJ/LAOBP sequences and the gap and secondary structural differences, including the Linker, in sequence alignment, both determine the subtle differences in protein 3D structure and the specificity of substrate recognition. Through GNM and ANM analysis, the three typical AABPs show similar motion tendency. In the open state without substrates, the receptor motion is highly coupled, with the Index Finger moving substantially and close to the Thumb, which facilitates substrate capture. With the association of substrates, the overall motility of AABPs is weakened, except for the C-termini interacting with the receptor on the inner membrane. Based on the 300 ns MD trajectories, two conserved key residues, 11 and 117, favoring receptor–ligand recognition, are obtained by energy decomposition, which may dominate the initiation of functional motion. Under the premise of rational division of AABPs’ functional domain and convergence of cMD trajectories, NRI-MD training provides a complete allosteric signal transmission process. Most of the allosteric signals are transduced within the Index Finger and are finally transmitted to the I-Loop region through global coordination, where there are certain crossings and overlaps conducive to signal enhancement and feedback regulation. The simulation results of this work are consistent with the previous experimental data, which not only reveals the allosteric mechanism of AABPs binding to specific amino acid substrates, but also provides theoretical support for the development of novel anticancer drugs based on functional regulation.

## Figures and Tables

**Figure 1 molecules-28-07139-f001:**
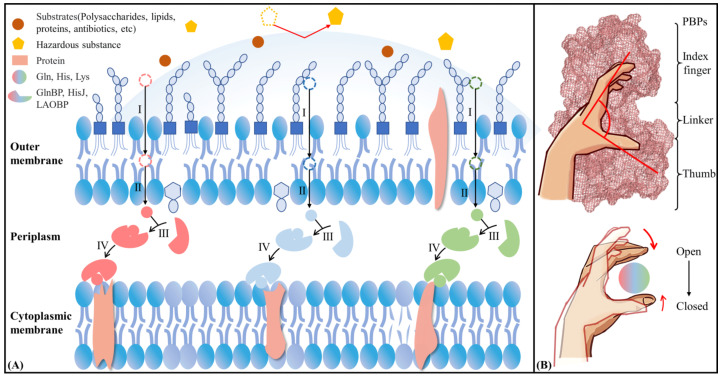
Transmembrane transport of nutrients (**A**) and allostery of periplasmic binding protein (**B**) in Gram-negative bacteria. Nutrients pass through the outer membrane of Gram-negative bacteria (I), enter the periplasmic space through channel proteins(II), are recognized by periplasmic binding proteins (III), and are transferred to the inner membrane (IV).

**Figure 2 molecules-28-07139-f002:**
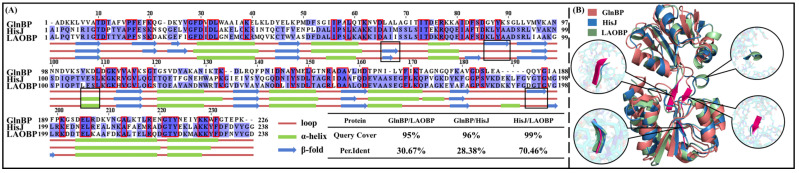
Sequence comparison and secondary structure comparison between the GlnBP, HisJ and LAOBP systems (**A**), red boxes indicate highly conserved residues in the three systems as well as comparison of their 3D structures. Light purple indicates that residues at the same location differ between protein sequences; Conversely, dark purple means that the residues are identical. (**B**). Black squares and round boxes represent regions with large secondary and 3D structural differences between GlnBP and other proteins, respectively. (Query cover: the degree of match between multiple sequences; Per.Ident: the consistency of the sequence alignment).

**Figure 3 molecules-28-07139-f003:**
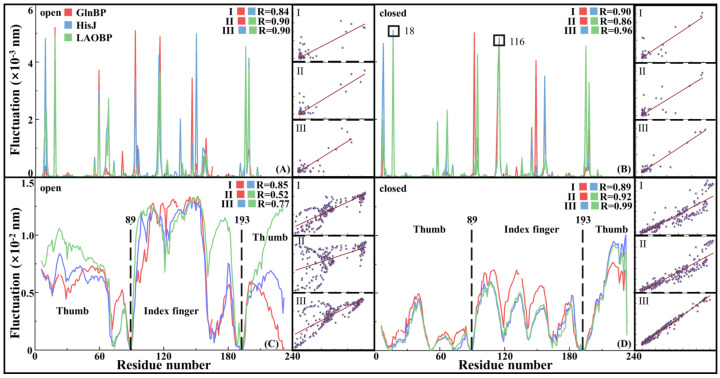
Comparison of fast (**A**,**B**) and slow (**C**,**D**) motion modes for the open (**A**,**C**) and closed (**B**,**D**) states.

**Figure 4 molecules-28-07139-f004:**
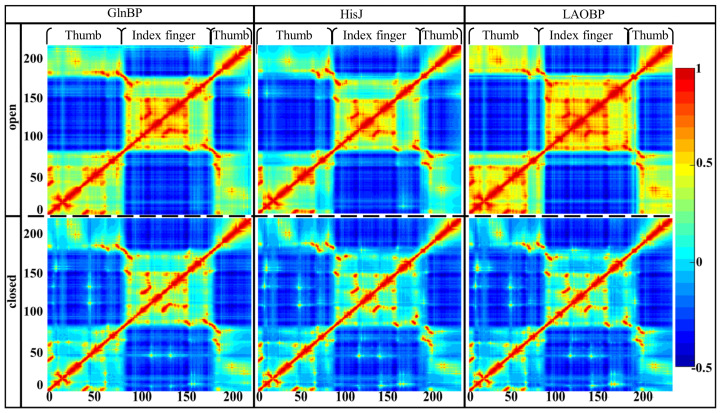
Motion correlation calculated by GNM for the GlnBP/HisJ/LAOBP monomers and substrate-bound complexes. Correlation values range from −0.5 to 1, where warm and cool colors represent the positive and negative correlations, respectively.

**Figure 5 molecules-28-07139-f005:**
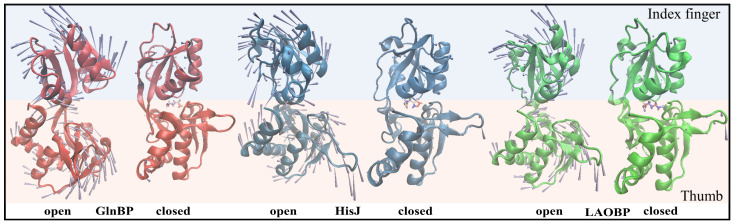
Functional motion patterns for the open and closed states of GlnBP/HisJ/LAOBP via ANM analysis. The length and direction of cone are used to characterize motion amplitude and direction, respectively. Light blue and light orange for Index Finger and Thumb region, respectively. Red, blue, and green respectively represent GlnBP, HisJ and LAOBP.

**Figure 6 molecules-28-07139-f006:**
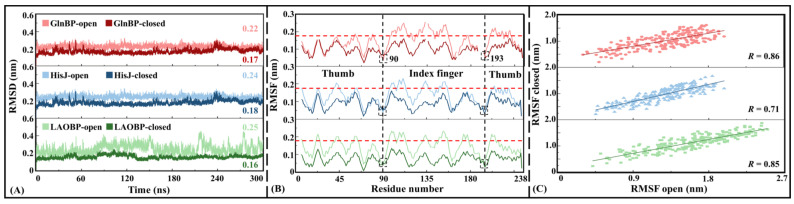
Convergence analysis of cMD trajectories for the GlnBP-open/GlnBP-closed, HisJ-open/HisJ-closed and LAOBP-open/LAOBP-closed systems. (**A**) RMSD distribution versus simulation time; (**B**) RMSF distribution at residual level; (**C**) The correlations of RMSF values between the closed and open states.

**Figure 7 molecules-28-07139-f007:**
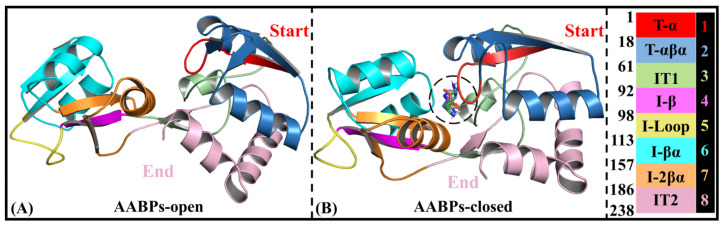
Domain partitions of the AABPs’ open (**A**) and closed (**B**) states, including eight parts, T-α, T-αβα, IT1, I-β, I-Loop, I-βα, I-2βα and IT2. For the convenience of subsequent domain interaction analysis, the last column of numbers 1–8 is used to represent the eight parts mentioned above.

**Figure 8 molecules-28-07139-f008:**
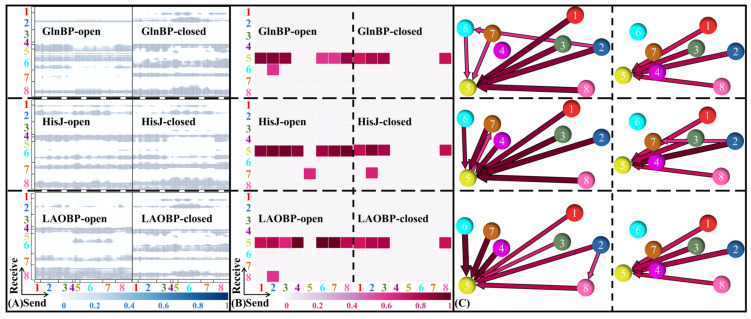
Distribution of residue interactions among learned edges (**A**)/domains (**B**) in six AABPs cMD simulations. (**C**) The interaction graph is mapped from the learned edges, where node thickness and arrow direction respectively indicate interaction strength and the direction of signal transmission. 1. T-α; 2. T-αβα; 3. IT1; 4. I-β; 5. I -Loop; 6. I-βα; 7. I-2βα; 8. IT2.

**Figure 9 molecules-28-07139-f009:**
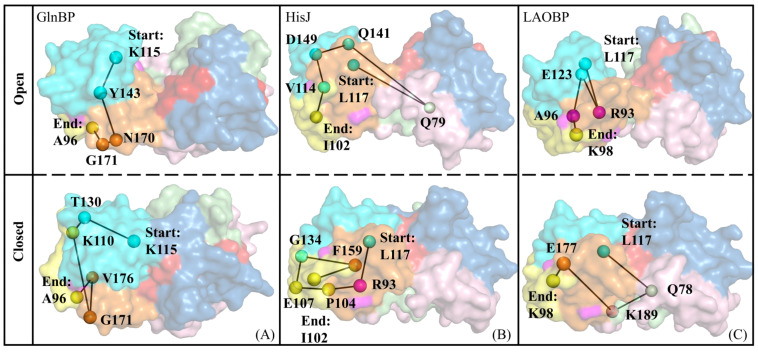
The most likely pathways mediating inter-domain allosteric communications in AABPs. GlnBP (**A**), HisJ (**B**), and LAOBP (**C**) mediate interdomain signal communication pathways in the open and closed states.

**Figure 10 molecules-28-07139-f010:**
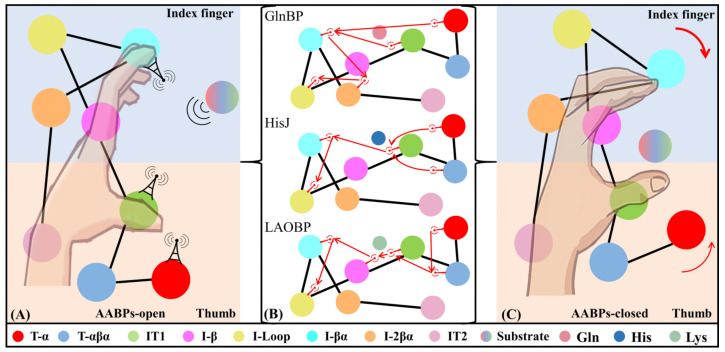
Possible allosteric mechanisms of AABPs. (**A**) The initial state of the AABPs’ open and closed substrate systems; (**B**) signaling between protein domains; (**C**) the closure of active pocket in the AABPs’ closed states.

**Table 1 molecules-28-07139-t001:** Key residues favoring the association of AABPs with substrates obtained by energy decomposition method.

Systems	Residues	*E_VDW_* ^a^	*E_ELE_* ^b^	*E_GB_* ^c^	*E_GBSUR_* ^d^	*E_TOT_*
GlnBP	117 (K115)	−0.44 ± 0.31	−6.97 ± 4.61	1.33 ± 3.8	−0.02 ± 0.02	−6.10 ± 0.52
	161 (D157)	−0.69 ± 0.8	−3.89 ± 3.44	2.57 ± 0.19	−0.04 ± 0.02	−2.04 ± 1.35
	69 (A67)	−0.23 ± 0.33	−2.92 ± 0.37	1.53 ± 0.34	−0.02 ± 0.01	−1.64 ± 0.39
	11 (D10)	−0.45 ± 0.01	−2.70 ± 0.22	1.72 ± 0.22	0.01 ± 0.00	−1.43 ± 0.01
	70 (G68)	−0.54 ± 0.01	−1.99 ± 1.32	1.24 ± 0.37	−0.03 ± 0.01	−1.33 ± 0.45
	121 (G119)	−0.22 ± 0.13	−1.61 ± 0.26	0.75 ± 0.25	−0.06 ± 0.01	−1.14 ± 0.12
HisJ	11 (D11)	0.11 ± 0.09	−9.80 ± 0.45	5.11 ± 0.56	−0.05 ± 0	−4.63 ± 0.02
	117 (L117)	−1.44 ± 0.06	−1.24 ± 1.70	0.90 ± 0.02	−0.07 ± 0.03	−1.85 ± 0.38
	69 (S69)	−0.37 ± 0.41	−1.13 ± 0.91	−0.14 ± 0.29	−0.05 ± 0.02	−1.68 ± 0.23
	14 (Y14)	−1.26 ± 0.09	−0.56 ± 0.90	0.43 ± 0.36	−0.08 ± 0.02	−1.45 ± 0.66
	52 (L52)	−0.95 ± 0.19	−0.09 ± 0.02	0.16 ± 0.01	−0.12 ± 0.01	−1.01 ± 0.17
LAOBP	117 (L117)	−0.91 ± 0	0.40 ± 0.33	−1.64 ± 0.14	−0.08 ± 0.01	−2.26 ± 0.16
	11 (D11)	−0.59 ± 0.18	−1.73 ± 0.04	0.21 ± 0.06	0.05 ± 0.02	−2.11 ± 0.23
	52 (F52)	−1.5 ± 0.07	−0.57 ± 0.04	0.17 ± 0.08	−0.07 ± 0.01	−1.98 ± 0.01
	72 (S72)	−1.17 ± 0.03	−0.86 ± 0.50	0.11 ± 0.04	−0.03 ± 0	−1.94 ± 0.09
	77 (R77)	−0.06 ± 0.04	−1.08 ± 0.50	−0.41 ± 0.02	−0.01 ± 0.01	−1.56 ± 0.16
	70 (S70)	−1.12 ± 0.08	−0.38 ± 0.01	0.40 ± 0.01	0.06 ± 0.02	−1.04 ± 0.26

^a^ *E_VDW_* and ^b^
*E_ELE_* are the van der Waals and electrostatic binding energies under vacuum, respectively. ^c^
*E_GB_* and ^d^
*E_GBSUR_* are the polar and non-polar parts of solvation effect, respectively. The unit of all data is kJ/mol.

## Data Availability

Not applicable.

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
