# Peer review of "Revealing Allosteric Mechanism of Amino Acid Binding Proteins from Open to Closed State"

_molecules, 2023, doi:10.3390/molecules28207139_

Round 1
Reviewer 1 Report
Hu and coworkers studied the allosteric mechanism of representative AABPs with elastic network models and MD simulations. The experiments and simulations were well-designed and thorough, providing many fine details about the protein's structural dynamics. As such, their results yielded new insights into the allosteric mechanism, including key residues that control the substrate binding and participate in domain motions. I believe that this is a significant work and is worthy of Molecules.
A few minor points:
1. In Figure 6a, the LAOB-open seems ‘noisy’ compared to other traces. Is this related to specific structural motions/factors?
2. Would the authors be able to speculate the timescales of the protein motions investigated in this study (without running more QM calculations/simulations)?
3. In Figure 10, the authors may want to introduce the color coding of the protein parts again to help the readers better understand the proposed mechanism.
4. Figure 5: It looks like the background colors (blue for index finger and yellow for thumb) are in the front, and the actual protein structures are a little blurry. Is this intentional?
Author Response
Prof. Jianping Hu, PhD
College of Pharmacy, Chengdu University
610106 Chengdu, China
Tel: +86-28-84616301
E-mail: [email protected]
Chengdu, September 12th, 2023
Dear Editorial Office
Editor, Molecules
Thank you very much for your email of Minor Revisions, regarding our manuscript Revealing allosteric mechanism of amino acid-binding proteins from open to closed state (Manuscript ID: molecules-2590979).
We would like to thank the three referees for their valuable and helpful comments about our manuscript. Reviewer 1 thinks we studied the allosteric mechanism of representative AABPs with elastic network models and MD simulations. The experiments and simulations were well-designed and thorough, providing many fine details about the protein's structural dynamics. As such, our results yielded new insights into the allosteric mechanism, including key residues that control the substrate binding and participate in domain motions. He believes that this is a significant work and is worthy of Molecules. Reviewer 2 states that we used multiple types of computational analysis to investigate AABP. GNM and ANM indicate AABP’s index finger and loop region move to a closed position on binding an amino acid ligand. cMD simulations indicate the index finger region and especially the I-Loop region have flexibility that suggests residues 11 and 117 are important for ligand recognition and receptor allostery. Finally, NRI-MD reveal information on AABP’s closure. The work may aid in the design of inhibitors of AABP ligand binding. Reviewer 3 considers The paper by us presents an interesting study of the mechanism of allosteric recognition in amino acid-binding proteins (AABPs) focusing on three examples (glutamine-binding protein, histidine-binding protein, and lysine/arginine/ornithine-binding protein) in E.coli. Using theoretical models (Gaussian network model, anisotropic network model) and performing molecular dynamics simulations, we identify the rigid and flexible regions of the investigated proteins as well as key residues of ligand recognition and signal transmission in AABPs. We made some good suggestions the changes have been implemented in our new version of the manuscript with red color. The changes have been implemented in our new version of the manuscript with red color. Our replies to reviewers’ comments are enclosed below and written in red. We hope that our revised manuscript is now suitable for publication in Molecules.
Looking forward to hearing from you.
Sincerely yours,
Jianping Hu, on behalf of all authors
Professor at College of Pharmacy, Chengdu University, 610106 Chengdu, China
Director of Drug Design and Molecular Simulation (DDMS) in Key Laboratory of Medicinal and Edible Plants Resources Development, Chengdu University
Specific comment 1:
In Figure 6a, the LAOBP-open seems ‘noisy’ compared to other traces. Is this related to specific structural motions/factors?
Response 1:
We thank the reviewer for raising this interesting question. The RMSD values reflect the phenomena that the individual partial atoms deviate from the average position - the motion magnitude of the individual atoms. The magnitude of change in RMSD reflects the overall flexibility and stability: the more stable the RMSD value, the more credible the simulated system is. In our system, LAOBP-open shows ‘noisy’ compared to other systems, but we think it's reasonable.The reasons are as follow: 1. As can be seen in Fig. 2A, LAOBP has more loop regions than other proteins, especially at residues D11-F29 and the C-terminal tail. As we all know, loop regions is more moveability than other second structure, introducing a ‘noisy’. 2. As seen in slow-motion changes analyzed using GNM (Fig 3C), the D11-F29 and C-terminal tails of LAOBP-open are more motile, accelerating the ‘noisy’ forming. 3. In order to further confirm whether these loop regions are the main factor for ‘noisy’ generation, we also examined the intermediate structure of the simulated process. By monitoring the simulation process with VMD, it was confirmed that the main cause of ‘noisy’ generation is the motion of the loop regions nor active regions. To sum up, the noise is not related to specific structural motions/factors, but only relationship with the motion of loop regions and C-terminal.
Figure 2. Sequence comparison and secondary structure comparison between the GlnBP, HisJ, and LAOBP systems (A), red boxes indicate highly conserved residues in the three systems; as well as comparison of their 3D structures (B). Black squares and round boxes represent regions with large secondary and 3D structural differences between GlnBP and other proteins, respectively. Query cover refers to the degree of match between multiple sequences (coverage). Per. Ident is the consistency of the sequence alignment.
Figure 3. Comparison of fast (A, B) and slow (C, D) motion modes for the open (A, C) and closed (B, D) states.
Specific comment 2:
Would the authors be able to speculate the timescales of the protein motions investigated in this study (without running more QM calculations/simulations)?
Response 2:
We also thank the reviewer for pointing out such a subtle problem. Quantum mechanics (QM) is a theory that studies the laws of motion of microscopic particles. The motion state of the particles is depicted by the wave function, the change law of the wave function is determined by the Schrödinger equation, which calculate the physical quantities. However, the longest detection time for QM is one-tenth of a nanosecond [1], while MD is now on the order of ns. In our work, we mainly want to study the overall conformational changes of the protein (MD simulation, GNM, ANM, NRI-MD), not the local conformational changes (QM). Therefore, QM technique is not the first choice for this work. Through combining MD simulation, GNM and NRI-MD technique, we studied the small-amplitude movements of the protein from the open state and the closed state, respectively, in order to obtain the main signaling pathways, which can help to reveal the mechanism of the conformational changes of AABPs binding to specific amino acid substrates. Additionally, we utilized ANM and key residue decomposition to further corroborate the results. On the time scale, we studied the overall movement over a short period of time (ns level) rather than the local dynamics of the active region, so we did not combine QM techniques to further analyze it.
Specific comment 3:
In Figure 10, the authors may want to introduce the color coding of the protein parts again to help the readers better understand the proposed mechanism.
Response 3:
Thank you for pointing out this question and we agree with it. In Figure 10, we would like to combine the global/ local conformational changes and molecular recognition analysis to propose a possible mechanism for the metastructuring of AABPs. To maintain consistency with the previous paper and to facilitate readers' understanding of the whole process studied in this work, we therefore color-coordinated the structures of the proteins in Fig. 10 according to Fig. 7. Under the reviewer's comments we found that we neglected the legend of Figure 10, which may cause an comprehension problem for readers, so in the revised manuscript we improved the addition of the legend of Figure 10, which is described below:
We have redrawn Figure 10 to include a legend to make it easier for the reader to understand the mechanism of metamorphosis between different regions of AAPBs.
Figure 10. Possible allosteric mechanisms of AABPs. (A) The initial state of the AABPs-open and substrate systems; (B) signaling between protein domains; (C) the closure of active pocket in the AABPs-closed states.
Specific comment 4:
Figure 5: It looks like the background colors (blue for index finger and yellow for thumb) are in the front, and the actual protein structures are a little blurry. Is this intentional?
Response 4:
Thanks for pointing out this question. In this figure we originally wanted to present the systems’ movement trend in open and closed states, in order to better distinguish the structural domains. From the figure, we used light blue and light orange to represent the index finger and thumb region respectively, but we uses a low transparency background color, which led a color overlaying on the proteins making the main display part of this figure become very blurred. So, in our revised manuscript, we first increase the transparency of the background color and improving image resolution; Increase the font size in the figure. The specific corrections are as follows:
Figure 5. Functional motion patterns for the open and closed states of GlnBP/ HisJ/ LAOBP via ANM analysis. The length and direction of cone are used to characterize motion amplitude and direction respectively. Light blue and light orange for Index finger and Thumb region, respectively.
Reference
[1] Wallrapp, F.H. and Guallar, V. Mixed quantum mechanics and molecular mechanics methods: Looking inside proteins. Wires. Comput. Mol. Sci. 2011, 1(2), 315-322; DOI: 10.1002/wcms.27.

Reviewer 2 Report
The authors used multiple type of computational analysis to investigate AABP. GNM and ANM indicate AABP’s index finger and loop region move to a closed position on binding an amino acid ligand. cMD simulations indicate the index finger region and especially the I-Loop region have flexibility that suggests residues 11 and 117 are important for ligand recognition and receptor allostery. Finally, NRI-MD reveal information on AABP’s closure. The work may aide in the design of inhibitors of AABP ligand binding.
Major concerns:
I have a concern with the nomenclature used to abbreviate the names of the binding proteins and the complete lack of basic biochemistry this shows. The correct single letter abbreviation for the amino acid should be used in the abriviated AABP name, for example: “lysine/arginine/ornithine binding protein (LAOBP)” – is this really the abbreviation decided on by the field for this protein, LAOBP would suggest to me leucine, alanine, ornithine, while KROBP would suggest lysine, arginine, ornithine
Author Response
Prof. Jianping Hu, PhD
College of Pharmacy, Chengdu University
610106 Chengdu, China
Tel: +86-28-84616301
E-mail: [email protected]
Chengdu, September 12th, 2023
Dear Editorial Office
Editor, Molecules
Thank you very much for your email of Minor Revisions, regarding our manuscript Revealing allosteric mechanism of amino acid-binding proteins from open to closed state (Manuscript ID: molecules-2590979).
We would like to thank the three referees for their valuable and helpful comments about our manuscript. Reviewer 1 thinks we studied the allosteric mechanism of representative AABPs with elastic network models and MD simulations. The experiments and simulations were well-designed and thorough, providing many fine details about the protein's structural dynamics. As such, our results yielded new insights into the allosteric mechanism, including key residues that control the substrate binding and participate in domain motions. He believes that this is a significant work and is worthy of Molecules. Reviewer 2 states that we used multiple types of computational analysis to investigate AABP. GNM and ANM indicate AABP’s index finger and loop region move to a closed position on binding an amino acid ligand. cMD simulations indicate the index finger region and especially the I-Loop region have flexibility that suggests residues 11 and 117 are important for ligand recognition and receptor allostery. Finally, NRI-MD reveal information on AABP’s closure. The work may aid in the design of inhibitors of AABP ligand binding. Reviewer 3 considers The paper by us presents an interesting study of the mechanism of allosteric recognition in amino acid-binding proteins (AABPs) focusing on three examples (glutamine-binding protein, histidine-binding protein, and lysine/arginine/ornithine-binding protein) in E.coli. Using theoretical models (Gaussian network model, anisotropic network model) and performing molecular dynamics simulations, we identify the rigid and flexible regions of the investigated proteins as well as key residues of ligand recognition and signal transmission in AABPs. We made some good suggestions the changes have been implemented in our new version of the manuscript with red color. The changes have been implemented in our new version of the manuscript with red color. Our replies to reviewers’ comments are enclosed below and written in red. We hope that our revised manuscript is now suitable for publication in Molecules.
Looking forward to hearing from you.
Sincerely yours,
Jianping Hu, on behalf of all authors
Professor at College of Pharmacy, Chengdu University, 610106 Chengdu, China
Director of Drug Design and Molecular Simulation (DDMS) in Key Laboratory of Medicinal and Edible Plants Resources Development, Chengdu University
Specific comment 1:
I have a concern with the nomenclature used to abbreviate the names of the binding proteins and the complete lack of basic biochemistry this shows. The correct single letter abbreviation for the amino acid should be used in the abbreviated AABP name, for example: “lysine/arginine/ornithine binding protein (LAOBP)” – is this really the abbreviation decided on by the field for this protein, LAOBP would suggest to me leucine, alanine, ornithine, while KROBP would suggest lysine, arginine, ornithine
Response 1:
We thank the reviewer for pointing out such a subtle problem. We paid attention to the single-letter abbreviation problem of amino acids when we learned the system through reading literature in the early stage: the names nomenclature of lysine/ arginine/ ornithine binding protein in the previous work was named by acronyms. For example, when Pang et al. studied AABP by molecular dynamics, the nomenclature for the lysine/ arginine/ ornithine binding protein was LAOBP[1]; the same nomenclature was used in the article by Hall et al. characterizing the conformational state of proteins by norm frequency[2]; coincidentally, LAOBP was also mentioned in an article studying data from molecular dynamics simulations to identify the hierarchical structure of protein dynamic structural domains[3]; until 2018, the nomenclature for studies on lysine/ arginine/ ornithine binding protein was still LAOBP[4]. We considered the LROBP method, but when we searched the crystal structure in Protein Data Bank(https://www.pdbus.org/), we found that the abbreviation of lysine/ arginine/ ornithine binding protein is LAO, which cannot be searched by LRO, such as 2LAO, 1LST, and 6MKX et al. Considering the subsequent studies of this system by others, we continue to follow the LAO nomenclature used by the former.
Reference
[1] Pang, A., Y. Arinaminpathy, M.S.P. Sansom, et al. Comparative molecular dynamics—similar folds and similar motions? Proteins: Structure, Function, and Bioinformatics 2005, 61(4), 809-822; DOI: 10.1002/prot.20672.
[2] Hall, B.A., S.L. Kaye, A. Pang, et al. Characterization of protein conformational states by normal-mode frequencies. J. Am. Chem. Soc. 2007, 129(37), 11394-11401; DOI: 10.1021/ja071797y.
[3] Yesylevskyy, S.O. Identifying the hierarchy of dynamic domains in proteins using the data of molecular dynamics simulations. Protein Peptide Lett. 2010, 17(4), 507-516; DOI: 10.2174/092986610790963636.
[4] Banda‐Vázquez, J., S. Shanmugaratnam, R. Rodríguez‐Sotres, et al. Redesign of LAOBP to bind novel l‐amino acid ligands. Protein Sci. 2018, 27(5), 957-968; DOI: 10.1002/pro.3403.

Reviewer 3 Report
The paper by Shi et al. presents an interesting study of the mechanism of allosteric recognition in amino acid-binding proteins (AABPs) focusing on three examples (glutamine-binding protein, histidine-binding protein, and lysine/arginine/ornithine-binding protein) in E. coli. Using theoretical models (Gaussian network model, nisotropic network model) and performing molecular dynamics simulations, they identify the rigid and flexible regions of the investigated proteins as well as key residues of ligand recogition and signal transmission in AABPs. I have only a couple of points to improve the manuscript:
1.The size of Fig. 2, Fig. 3, and Fig. 5 should be increased for better viewing. In particular the letter size of the text in the figures should be increased.
2. The figure caption of Fig. 2 is not clear.
3. 300 ns MD simulation to address slow motions including domain motions and the correlation between them is quite short. Would the authors comment on this?
There are a few sentences in which the English could be improved.
e.g.
pg. 1, line 12: "significantly"
pg. 8, line 323: "Index finger exhibits more flexible than Thumb"
Author Response
Prof. Jianping Hu, PhD
College of Pharmacy, Chengdu University
610106 Chengdu, China
Tel: +86-28-84616301
E-mail: [email protected]
Chengdu, September 12th, 2023
Dear Editorial Office
Editor, Molecules
Thank you very much for your email of Minor Revisions, regarding our manuscript Revealing allosteric mechanism of amino acid-binding proteins from open to closed state (Manuscript ID: molecules-2590979).
We would like to thank the three referees for their valuable and helpful comments about our manuscript. Reviewer 1 thinks we studied the allosteric mechanism of representative AABPs with elastic network models and MD simulations. The experiments and simulations were well-designed and thorough, providing many fine details about the protein's structural dynamics. As such, our results yielded new insights into the allosteric mechanism, including key residues that control the substrate binding and participate in domain motions. He believes that this is a significant work and is worthy of Molecules. Reviewer 2 states that we used multiple types of computational analysis to investigate AABP. GNM and ANM indicate AABP’s index finger and loop region move to a closed position on binding an amino acid ligand. cMD simulations indicate the index finger region and especially the I-Loop region have flexibility that suggests residues 11 and 117 are important for ligand recognition and receptor allostery. Finally, NRI-MD reveal information on AABP’s closure. The work may aid in the design of inhibitors of AABP ligand binding. Reviewer 3 considers The paper by us presents an interesting study of the mechanism of allosteric recognition in amino acid-binding proteins (AABPs) focusing on three examples (glutamine-binding protein, histidine-binding protein, and lysine/arginine/ornithine-binding protein) in E.coli. Using theoretical models (Gaussian network model, anisotropic network model) and performing molecular dynamics simulations, we identify the rigid and flexible regions of the investigated proteins as well as key residues of ligand recognition and signal transmission in AABPs. We made some good suggestions the changes have been implemented in our new version of the manuscript with red color. The changes have been implemented in our new version of the manuscript with red color. Our replies to reviewers’ comments are enclosed below and written in red. We hope that our revised manuscript is now suitable for publication in Molecules.
Looking forward to hearing from you.
Sincerely yours,
Jianping Hu, on behalf of all authors
Professor at College of Pharmacy, Chengdu University, 610106 Chengdu, China
Director of Drug Design and Molecular Simulation (DDMS) in Key Laboratory of Medicinal and Edible Plants Resources Development, Chengdu University
Specific comment 1:
The size of Fig. 2, Fig. 3, and Fig. 5 should be increased for better viewing. In particular the letter size of the text in the figures should be increased.
Response 1:
We thank the reviewer for raising this interesting question. Improving the clarity of an image facilitates the reader's ability to read the content and have a better view. In our work, we originally used 300 dpi for the images, but after listening to the reviewer's comments, we decided to increase the resolution to 600 dpi. In addition, we also make changes to increase the font size and bolding of the fonts so as to ensure the clarity of the images and improve the reading experience of the readers. In addition to Figures 2, 3, and 5 mentioned by the reviewer, we also improved the other images. Including increase the font size and bolding the fonts in Figs. 2, 3, 5; adjusting the background color in Fig. 5 to improve the clarity of the proteins; and adding a legend in Fig. 10, so that readers can understand better. Below are the figures after our correction:
Figure 2. Sequence comparison and secondary structure comparison between the GlnBP, HisJ, and LAOBP systems (A), red boxes indicate highly conserved residues in the three systems; as well as comparison of their 3D structures (B). Black squares and round boxes represent regions with large secondary and 3D structural differences between GlnBP and other proteins, respectively. Query cover refers to the degree of match between multiple sequences (coverage). Per. Ident is the consistency of the sequence alignment.
Figure 3. Comparison of fast (A, B) and slow (C, D) motion modes for the open (A, C) and closed (B, D) states.
Figure 5. Functional motion patterns for the open and closed states of GlnBP/ HisJ/ LAOBP via ANM analysis. The length and direction of cone are used to characterize motion amplitude and direction respectively. Light blue and light orange for Index finger and Thumb region, respectively.
Figure 10. Possible allosteric mechanisms of AABPs. (A) The initial state of the AABPs-open and substrate systems; (B) signaling between protein domains; (C) the closure of active pocket in the AABPs-closed states.
Specific comment 2:
The figure caption of Fig. 2 is not clear.
Response 2:
We also thank the reviewer for pointing out such a subtle problem. The original write-up was written to express that we had organized the primary sequence, secondary structure and 3D structure of three AABPs. At the suggestion of the reviewer, we reviewed the content and found that the figure could have been written better, including highlighting the comparison between the structures of three AABPs; the meanings of the titles in the table; and the meanings of the red and black boxes in the figure. The specific changes are as follows:
Figure 2. Sequence comparison and secondary structure comparison between the GlnBP, HisJ, and LAOBP systems (A), red boxes indicate highly conserved residues in the three systems; as well as comparison of their 3D structures (B). Black squares and round boxes represent regions with large secondary and 3D structural differences between GlnBP and other proteins, respectively. (Query cover: the degree of match between multiple sequences; Per. Ident: the consistency of the sequence alignment).
Specific comment 3:
300 ns MD simulation to address slow motions including domain motions and the correlation between them is quite short. Would the authors comment on this?
Response 3:
Thank you for pointing out this problem and we agree with this comment. Theoretically, the simulation timescale is infinite, in practice it depends on your computing power and computational resources, as well as modeling accuracy. The smaller the modeling accuracy and the coarser the model, the larger the chosen time scale will be. The time scale of molecular dynamics simulation in current research is basically 100-500 ns. 300 ns is very common [1-5]: Miao et al. studied the accelerated dynamic simulation of protein folding by 300 ns in his study [1]; Zou et al. studied Norepinephrine inhibits Alzheimer's amyloid-β-phosphate by 300 ns dynamic simulation [2]. Therefore, the length of the simulation time is not the only criterion to ensure that the simulated system; it is the criterion to determine when the system reaches a stable structure. In our research system, it can be found that with 300 ns of simulation, the system has reached stability for a period of time, which is characterized as follows: 1. selecting models with high precision. The initial structures of the proteins we chose, including open and closed states, are derived from the protein database (https://www.pdbus.org/), and the high precision of the model reduces the time scale of the simulation; 2. Stability analysis. From Fig. 6A, it can be found that after 300 ns, the three systems remained stable for a long time without further substantial conformational changes. So, in our study, 300 ns MD simulation to address slow motions including domain motions and the correlation between them is enough.
Figure 6. Convergence analysis of cMD trajectories for the GlnBP-open/ GlnBP-closed, HisJ-open/ HisJ-closed, and LAOBP-open/ LAOBP-closed systems. (A) RMSD distribution versus simulation time; (B) RMSF distribution at residual level; (C) The correlations of RMSF values between the closed and open states.
Other comment:
- 1, line 12: "significantly"
- 8, line 323: "Index finger exhibits more flexibility than Thumb"
Response 3:
Thanks for pointing out this. Grammar is the most basic and equally important thing for an English article, grammar improves the reader's reading flow and makes it easier to understand the purpose and meaning conveyed by the text. The reviewer's suggestions can improve the readability and fluency of the article, and we are happy to accept them. In addition to the grammatical issues raised by the reviewers, we have further checked the grammar of the whole text and made changes in red color in the text. Below are some of the changes:
- 1, line 12: "significantly" is revised as “significant”.
- 8, line 323: “is” is revised as “remains”
- 8, line 323: “Index finger” is revised as “Index finger region”
- 8, line 323: “more flexible” is revised as “higher flexibility”
- 8, line 324: “Thumb” is revised as “Thumb region,”
- 10, line 395: “generally” is revised as “general”
- 11, line 432: “closted-state” is revised as “closed-state”
Reference
[1] Miao, Y., F. Feixas, C. Eun, et al. Accelerated molecular dynamics simulations of protein folding. J. Comput. Chem. 2015, 36(20), 1536-1549; DOI: 10.1002/jcc.23964.
[2] Zou, Y., Z. Qian, Y. Chen, et al. Norepinephrine inhibits Alzheimer’s amyloid-β peptide aggregation and destabilizes amyloid-β protofibrils: A molecular dynamics simulation study. ACS Chem. Neurosci. 2019, 10(3), 1585-1594; DOI: 10.1021/acschemneuro.8b00537.
[3] Marinelli, F., F. Pietrucci, A. Laio, et al. A kinetic model of trp-cage folding from multiple biased molecular dynamics simulations. PLoS Comput. Biol. 2009, 5(8), e1000452; DOI: 10.1371/journal.pcbi.1000452.
[4] Altis, A., P.H. Nguyen, R. Hegger, et al. Dihedral angle principal component analysis of molecular dynamics simulations. J. Chem. Phy. 2007, 126(24); DOI: 10.1063/1.2746330.
[5] Harmandaris, V.A., V.G. Mavrantzas, D.N. Theodorou, et al. Crossover from the rouse to the entangled polymer melt regime: signals from long, detailed atomistic molecular dynamics simulations, supported by rheological experiments. Macromolecules 2003, 36(4), 1376-1387; DOI: 10.1021/ma020009g.
